# Learning to Select the Best Forecasting Tasks for Clinical Outcome Prediction

**Yuan Xue**[*], **Nan Du**[*], **Anne Mottram, Martin Seneviratne, Andrew M. Dai**
Google
{yuanxue, dunan, annemottram, martsen, adai}@google.com

## Abstract

The paradigm of 'pretraining' from a set of relevant auxiliary tasks and then 'finetuning' on a target task has been successfully applied in many different domains. However, when the auxiliary tasks are abundant, with complex relationships to the target task, using domain knowledge or searching over all possible pretraining setups is inefficient and suboptimal. To address this challenge, we propose a method to automatically select from a large set of auxiliary tasks, which yields a representation most useful to the target task. In particular, we develop an efficient algorithm that uses automatic auxiliary task selection within a nested-loop meta-learning process. We have applied this algorithm to the task of clinical outcome predictions in electronic medical records, learning from a large number of self-supervised tasks related to forecasting patient trajectories. Experiments on a real clinical dataset demonstrate the superior predictive performance of our method compared to direct supervised learning, naive pretraining and simple multitask learning, in particular in low-data scenarios when the primary task has very few examples. With detailed ablation analysis, we further show that the selection rules are interpretable and able to generalize to unseen target tasks with new data.

## 1 Introduction

The wide adoption of electronic medical record (EMR) systems has generated large repositories of patient data in the form of multivariate time-series. These data are increasingly used for supervised learning, with the goal of providing decision support to clinicians by predicting clinical outcomes for individual patients [1]. Recent examples have focused on the prediction of inpatient mortality [2], acute kidney injury [3], circulatory shock [4], etc.

One major challenge with EMR modeling is that the raw data is high-dimensional, noisy, sparse and heterogeneous, as it is generated in the course of routine clinical care [5]. Furthermore, accurately labeling clinical endpoints can be extremely challenging and often requires time-consuming manual chart review by clinicians, meaning that modeling must be data efficient. Even in cases where the outcome label is more clearly encoded, e.g. mortality, data availability is often still an issue as there are only a limited number of patients with that outcome in a selected cohort.

To tackle these issues of data quality and label shortage, a common approach widely applied in computer vision (CV) and natural language processing (NLP) domains is pretraining and finetuning. Pretraining involves learning a compact representation on related tasks with abundant data. These learned representations can then be finetuned on the primary task with limited labels, assisting supervised performance by leveraging prior knowledge.

EMRs contain thousands of different laboratory tests, observations, medications, procedures *etc.*, for each patient over time. Using the trajectories of these time series data as self-supervised objectives provides a promising way to learn a useful patient representation. However, naively pretraining across

---

[*]Authors contributed equally.

all measurements can easily lead to a representation oblivious of the target clinical prediction task. Furthermore, pretraining from trivial and less important measurements may overshadow important signals in the learned representation. Since the number of available measurement trajectories is large, an exhaustive search over all possible task combinations is not tractable. With complex relationships between the measurement trajectories and the primary outcome, the decision of how to structure pretraining is not a straightforward process.

To address this challenge, the goal of this paper is to automatically select and mix the most relevant auxiliary tasks to be used in pretraining for a specific primary task. In particular, we introduce a new connection between multitask learning and transfer learning within the framework of meta learning. Each auxiliary task is a self-supervised trajectory forecast for a specific clinical measurement, and the primary target task involves supervised learning based on the learned representation. We propose an efficient gradient-based algorithm that learns to automatically select the most relevant auxiliary tasks for pretraining, and then optimizes the meta objective of generalizing to the target task.

Experiments on real world clinical datasets show that the learned representation from the selected auxiliary tasks leads to favorable predictive performance compared to both direct supervised learning, naive pretraining and simple multitask learning. This advantage further increases in low data regimes where the target task has few labeled examples. Detailed ablation analysis demonstrates that the selected auxiliary tasks are meaningful and able to generalize to unseen target tasks.

## 2 Learning Tasks

In a longitudinal EMR dataset, a patient's record is a collection of sequential clinical-visit data which can be naturally represented as multi-variate time series. Each time series captures the readings over time from one type of clinical measurement (e.g., blood pressure, lactate, etc.), or intervention (e.g., ventilator settings). For a given patient, we use $x_t^f$ to represent the $f$th feature value $f \in \mathcal{F}$ at the time step $t$. $\mathbf{x}_T^f = \{x_t^f\}_{t=1}^T$ denotes the $f$th time series, and $T$ is the number of time steps. We also use $\boldsymbol{x}_t$ as the $|\mathcal{F}|$-dimensional feature vector at the time $t$.

**Primary supervised task: Clinical outcome prediction.** For each sequence $\{\boldsymbol{x}_t\}_{t=1}^T$, there is an associated label $y$ representing the occurrence of a clinical outcome of interest, e.g., sepsis, shock, mortality, etc. The goal is to learn a model that predicts the most likely label value $\hat{y}$ for a given input sequence $\{\boldsymbol{x}_t\}_{t=1}^T$. The learning process thus takes the standard form of supervised learning with a loss $\ell(\hat{y}, y)$ associated with the model.

**Auxiliary task: Trajectory forecast.** The goal of the trajectory forecast task is to model the distribution of the future values of raw EMR data elements $p(\mathbf{x}_{\tau+1:\tau+H}^f | \mathbf{x}_{1:\tau}^f)$ given the past history $\mathbf{x}_{1:\tau}^f$. Here, $\tau$ is the time of prediction, $H$ represents the number of time steps we look into the future, and $\mathbf{x}_{\tau+1:\tau+H}^f = \{x_t^f\}_{t=\tau+1}^{\tau+H}$. This task by nature takes the form of self-supervised learning since the future values of a time series can be easily treated as the learning signal. Compared to the clinical outcome prediction task, the patient's trajectory forecast task requires no human labels, and many powerful self-supervised techniques can be applied to the task [6–13].

We can expect that pretraining with self-supervised trajectory forecast tasks for each feature $f \in \mathcal{F}$ will produce useful patient representations for the clinical outcome prediction task which often has few examples. However, when the set of auxiliary tasks $|\mathcal{F}|$ is large, both joint pretraining using all the tasks in $\mathcal{F}$, or successive pretraining in an iterative way, can be sub-optimal and inefficient in that not all the auxiliary tasks are equally useful for transferring knowledge to the target primary task, leading to a less informative representation for downstream tasks.

## 3 Automatic Task Selection

We study the problem of learning to select the most relevant trajectory forecast tasks so that the learned representation is optimized for improving the performance of the target clinical outcome prediction task (schematic in Figure 1). In the following sections, we present our problem formulation, model design, and learning algorithms.

### 3.1 Problem Formulation

Selecting the optimal subset of auxiliary trajectory forecast tasks from $\mathcal{F}$ requires exploring $2^{|\mathcal{F}|}$ combinations, which is prohibitive in practice. To make the search space continuous, we relax

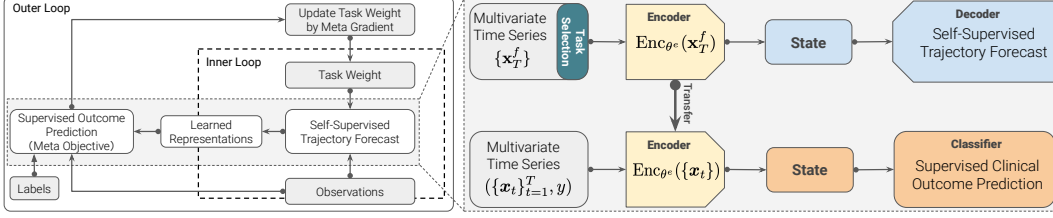

Figure 1: Schematic for guiding pretraining by supervised learning on a primary outcome via a nested-loop meta-learning process. The inner loop learning uses a sequence-to-sequence architecture for trajectory prediction. The utility of the learned representation from the encoder is measured by the supervised learning in the outer loop, and then used to update the weight of each trajectory prediction task in the inner loop via gradient descent.

the categorical choice of a particular task $f$ by learning a weight $\lambda_f$, $\sum_f \lambda_f = 1$ to indicate the importance of each task. Alternatively, $\lambda$ can be treated as a learned distribution over the task space $\mathcal{F}$. For each task $f \in \mathcal{F}$, without loss of generality, we use a very basic sequence to sequence model, which can easily be extended to more advanced attention-based models [7] or state-space models [8–13].

$$\mathbf{Enc}_{\theta^e}(\mathbf{x}_{1:\tau}^f) = s_\tau, \ \ \mathbf{Dec}_{\theta^d}(s_\tau) = \hat{\mathbf{x}}_{\tau+1:\tau+H}^f, \ \ \ell^p(\theta^e, \theta^d|\lambda) = \sum_f \lambda_f \|\hat{\mathbf{x}}_{\tau+1:\tau+H}^f - \mathbf{x}_{\tau+1:\tau+H}^f\|_2^2,$$

where the encoder $\mathbf{Enc}_{\theta^e}$ encodes the input sequence $\mathbf{x}_{1:\tau}^f$ up to time $\tau$ into an intermediate state $s_\tau$ from which the decoder $\mathbf{Dec}_{\theta^d}$ decodes the predicted sequence $\hat{\mathbf{x}}_{\tau+1:\tau+H}^f$. Here, we use the regression loss for simplicity. $\ell^p$ denotes the pretraining loss. Other reconstruction losses, *e.g.*, negative log likelihood losses, can also be easily applied. For the primary supervised learning task, we reuse the pretrained encoder, and train an additional classifier head for outcome prediction with a typical cross-entropy log loss

$$\mathbf{Enc}_{\theta^e}(\{\boldsymbol{x}_{1:\tau}\}) = h_\tau, \quad \ell^c(\theta^e, \theta^c) = \log p_{\theta^c}(y|h_\tau), \tag{1}$$

where $\theta^c$ are the parameters of the classifier, and $(\{\boldsymbol{x}_{1:\tau}\}, y)$ is the training example of one patient, and $\ell^c$ denotes the classification loss.. We formulate the task selection problem into the following optimization task.

$$\begin{aligned}
&\underset{\lambda}{\text{minimize}}\ \ell_{\text{val}}^c\big(\lambda|\theta_{N_S}^e, \theta_{N_S}^c\big) \text{ such that} \\
&\theta_{N_S}^e, \theta_{N_S}^c = \operatorname{argmin} \ell_{\text{train}}^c(\theta^e, \theta^c), \ \ \theta_{N_S}^e = \Phi_{\text{train}}(\theta_{N_P}^e) \rightarrow \text{supervised learning} \\
&\theta_{N_P}^e, \theta_{N_P}^d = \underset{\theta^e, \theta^d}{\operatorname{argmin}} \ell_{\text{train}}^p(\theta^e, \theta^d|\lambda) \rightarrow \text{self-supervised learning}
\end{aligned} \tag{2}$$

where $N_P$ and $N_S$ are the number of training update steps of the **P**retraining and the **S**upervised training, respectively. $\theta_{N_P}^e$ and $\theta_{N_P}^d$ are the encoder and decoder parameters learned after $N_P$ steps of self-supervised training. We then switch to the target task by treating $\theta_{N_P}^e$ as the initialization of the encoder. $\Phi_{\text{train}}(\theta_{N_P}^e)$ is an operator that updates the initialization $\theta_{N_P}^e$ for $N_S$ steps to obtain $\theta_{N_S}^e$. In practice, $\Phi_{\text{train}}$ can be the normal gradient update using the train data of the target task. Here, we explicitly use $\Phi_{\text{train}}$ to denote the initialization from $\theta_{N_P}^e$. More importantly, because the self-supervised learning output $\theta_{N_P}^e$ implicitly depends on a given $\lambda$, it further enables the learning of $\lambda$ using the error signals back-propagated from the target task. That is, we explicitly optimize the validation loss $\ell_{\text{val}}^c\big(\lambda|\theta_{N_S}^e, \theta_{N_S}^c\big)$ of the target task, which is often referred to as the response function (or meta objective), with respect to $\lambda$ known as the hyper (or meta) parameter, so that the generalization of the target task can be directly optimized.

### 3.2 Bi-level Optimization

The optimization of the meta objective 2 determines the quality of the learned representation from the encoder with parameter $\theta_{N_p}^e$, and it includes two loops of learning processes shown in Figure 1. Given a fixed $\lambda$ as one configuration, the inner loop first finds a candidate representation through the self-supervised learning on the trajectory forecast tasks, some of which receives more attention while others may be discarded. It then finetunes the representation via supervised learning on the clinical outcome prediction task. The quality of the learned representation is measured by the meta objective of the outer loop. The outer loop then updates the configuration of the inner loop to locate a

potentially better hypothesis space where the two objectives $\ell^p$ and $\ell^c$ will be minimized again. This nested learning process also arises in gradient-based hyper-parameter optimization[14, 15], and can be reformulated as follows.

$$\underset{\lambda,\{\theta_i^e\},\{\theta_i^d\},\{\theta_i^c\}}{\text{minimize}} \ell_{\text{val}}^c(\theta_{N_S}^e, \theta_{N_S}^c) \text{ such that}$$

$$\theta_i^e = \Psi_i^e(\theta_{i-1}^e, \theta_{i-1}^d, \lambda), \theta_i^d = \Psi_i^d(\theta_{i-1}^e, \theta_{i-1}^d, \lambda), i \in [1, N_P],$$

$$\theta_i^e = \Phi_i^e(\theta_{i-1}^e, \theta_{i-1}^c), \theta_i^c = \Phi_i^c(\theta_{i-1}^e, \theta_{i-1}^c), i \in [N_P + 1, N_P + N_S], \tag{3}$$

where $\Psi_i$ and $\Phi_i$ represent the gradient step of the optimization that updates the parameters at step $i$ in the respective pretrain and finetune stage. This reformulates the implicit dependencies among the parameters in the training procedure into explicit optimization constraints. The Lagrangian of problem 3 is thus

$$\mathcal{L}(\lambda, \{\theta_i^e\}, \{\theta_i^d\}, \{\theta_i^c\}, \alpha, \beta, \gamma, \delta) = \ell_{\text{val}}^c(\theta_{N_S}^e, \theta_{N_S}^c) + \sum_{i=1}^{N_P} \alpha_i \left(\Psi_i^e(\theta_{i-1}^e, \theta_{i-1}^d, \lambda) - \theta_i^e\right) + \tag{4}$$

$$\sum_{i=1}^{N_P} \gamma_i \left(\Psi_i^d(\theta_{i-1}^e, \theta_{i-1}^d, \lambda) - \theta_i^d\right) + \sum_{i=N_P+1}^{N_P+N_S} \beta_i \left(\Phi_i^e(\theta_{i-1}^e, \theta_{i-1}^c) - \theta_i^e\right) + \delta_i \left(\Phi_i^c(\theta_{i-1}^e, \theta_{i-1}^c) - \theta_i^c\right)$$

where for each step $i$, $\alpha_i$, $\beta_i$, $\gamma_i$, and $\delta_i$ are the associated row vectors of Lagrangian multipliers. Since the encoder parameter $\theta_{N_p}^e$ at the last step of the pretraining builds the connection between the self-supervised trajectory forecast tasks and the supervised clinical outcome prediction task, and the value at the last supervised step $\theta_{N_s}^e$ is used for predictions, their derivatives are first given by

$$\nabla_{\theta_{N_P}^e} \mathcal{L} = -\alpha_{N_P} + \beta_{N_P+1} \left(\nabla_{\theta_{N_P}^e} \Phi_{N_P+1}^e(\theta_{N_P}^e, \theta_{N_P}^c, \lambda)\right) \tag{5}$$

$$\nabla_{\theta_{N_S}^e} \mathcal{L} = \nabla_{\theta_{N_S}^e} \ell_{\text{val}}^c - \beta_{N_S}, \nabla_{\theta_{N_P}^d} \mathcal{L} = -\gamma_{N_P} \tag{6}$$

Then, at each intermediate step $i$ in the pretrain and finetune stage, the respective derivatives are

$$\nabla_{\theta_i^e} \mathcal{L} = -\alpha_i + \alpha_{i+1} \nabla_{\theta_i^e} \Psi_{i+1}^e(\theta_i^e, \theta_i^d, \lambda), \text{ for } i \in [1, N_P - 1],$$

$$\nabla_{\theta_i^e} \mathcal{L} = -\beta_i + \beta_{i+1} \nabla_{\theta_i^e} \Phi_{i+1}^e(\theta_i^e, \theta_i^c), \text{ for } i \in [N_P + 1, N_P + N_S - 1],$$

$$\nabla_{\theta_i^d} \mathcal{L} = -\gamma_i + \gamma_{i+1} \nabla_{\theta_i^d} \Psi_{i+1}^d(\theta_i^e, \theta_i^d, \lambda), \text{ for } i \in [1, N_P - 1]. \tag{7}$$

Finally, we can derive the gradient of the hyper-parameter $\lambda$ as

$$\nabla_\lambda \mathcal{L} = \sum_{i=1}^{N_P} \left(\alpha_i \nabla_\lambda \Psi_i^e(\theta_{i-1}^e, \theta_{i-1}^d, \lambda) + \gamma_i \nabla_\lambda \Psi_i^d(\theta_{i-1}^e, \theta_{i-1}^d, \lambda)\right). \tag{8}$$

The optimal conditions are then obtained by setting each derivative to zero.

$$\beta_{N_S} = \nabla_{\theta_{N_S}^e} \ell_{\text{val}}^c(\theta_{N_S}^e, \theta_{N_S}^c) \rightarrow \text{supervised objective} \tag{9}$$

$$\beta_i = \beta_{i+1} \nabla_{\theta_i^e} \Phi_{i+1}^e(\theta_i^e, \theta_i^c), \; i \in [N_P + 1, N_P + N_S - 1] \rightarrow \text{supervised learning} \tag{10}$$

$$\alpha_{N_P} = \beta_{N_P+1} \nabla_{\theta_{N_P}^e} \Phi_{N_P+1}^e(\theta_{N_P}^e, \theta_{N_P}^c) \rightarrow \text{knowledge transfer} \tag{11}$$

$$\alpha_i = \alpha_{i+1} \nabla_{\theta_i^e} \Psi_{i+1}^e(\theta_i^e, \theta_i^d, \lambda), \; i \in [1, N_P - 1] \rightarrow \text{self-supervised learning} \tag{12}$$

We first observe that Equation 10 back-propagates the signal from the meta-objective that quantifies the utility of the learned representation from the encoder through the supervised learning process. Equation 11 is the *touching point* of the two learning processes that further transfers this signal back to the self-supervised learning stage. Finally, Equation 12 distributes the signal to each learning step of the pretraining process. Compared to the encoder that is involved in both pretraining and finetuning, the decoder of the sequence to sequence model is only used in the pretraining stage to serve the self-supervised loss only. As a result, even though the decoder $\theta_i^d$ is involved in $\nabla_\lambda \Psi_i^e$ to measure how fast the gradient of the encoder can change w.r.t $\lambda$, the second order information $\nabla_\lambda \Psi_i^d$ from the decoder itself is not needed to update $\lambda$. This is also verified by the optimality condition that $\gamma_{N_P} = 0$ and $\gamma_i = \gamma_{i+1} \nabla_{\theta_i^d} \Psi_{i+1}^d(\theta_i^e, \theta_i^d, \lambda)$ from Equation 7. Therefore, the gradient of $\lambda$ can be solely determined by the signals of both $\alpha$ and $\beta$ from Equation 10 to 12. The full algorithm is given in the Appendix.

**Algorithm 1:** First-Order Automatic Task Selection

---

Randomly initialize $\theta_0^e$, $\theta_0^d$, $\theta_{N_P}^c$ and $\lambda$;

**for** $k = 1, 2, ...$ **do**

  **for** $i \in [1, N_P]$ **do**                                ▷ `Self-supervised learning loop`

      $\theta_i^e = \Psi_i^e \left( \theta_{i-1}^e, \theta_{i-1}^d, \lambda_k \right), \theta_i^d = \Psi_i^d \left( \theta_{i-1}^e, \theta_{i-1}^d, \lambda_k \right)$;          ▷ `Gradient descent`

  Get $a = \partial \ell_{\text{train}}^p / \partial \theta_{N_P}^e$, and $b = \partial \ell_{\text{train}}^p / \partial \lambda$;

  **for** $i \in [N_P + 1, N_P + N_S]$ **do**                    ▷ `Supervised learning loop`

      $\theta_i^e = \Phi_i^e \left( \theta_{i-1}^e, \theta_{i-1}^c \right), \theta_i^c = \Phi_i^c \left( \theta_{i-1}^e, \theta_{i-1}^c \right)$;            ▷ `Gradient descent`

  Get $c = \partial \ell_{\text{val}}^c / \partial \theta_{N_S}^e$;

  Get $g_\lambda = c \cdot (1/a) \cdot b$;                  ▷ `Compute hyper-gradient by Equation` 13

  $\lambda_k = \lambda_{k-1} - \epsilon \cdot g_\lambda$;                        ▷ `Gradient descent`

**return** $\theta_{N_P}^e$, $\lambda$

---

### 3.3 Efficient Gradient-based Learning Algorithm

Exact evaluation of Equation 8 is expensive in that $\nabla_\lambda \Psi_i^e(\theta_{i-1}^e, \theta_{i-1}^d, \lambda)$ and $\nabla_{\theta_i^e} \Phi_{i+1}^e(\theta_i^e, \theta_i^c)$ include the Jacobian and Hessian matrix of the gradient update operation $\Psi_i^e$ and $\Phi_{i+1}^e$. Motivated by related techniques in [16], we propose an efficient first-order approximation to Equation 8. More specifically, given that $\theta_{N_S}^e = \Phi_{\text{train}}(\theta_{N_P}^e) = \theta_{N_P}^e + \sum_{i=N_P}^{N_P + N_S - 1} \nabla_{\theta_i^e} \ell_{\text{train}}^c$ in Equation 2, the gradients $\left\{ \nabla_{\theta_i^e} \ell_{\text{train}}^c \right\}$ are treated as constants [16]. By applying the chain rule with the gradient approximation, we can have

$$\frac{\partial \ell_{\text{val}}^c}{\partial \lambda} = \frac{\partial \ell_{\text{val}}^c \left( \theta_{N_S}^e, \theta_{N_S}^c \right)}{\partial \theta_{N_S}^e} \cdot \frac{\partial \theta_{N_S}^e}{\partial \theta_{N_P}^e} \cdot \frac{\partial \theta_{N_P}^e}{\partial \ell_{\text{train}}^p(\theta^e, \theta^d | \lambda)} \cdot \frac{\partial \ell_{\text{train}}^p(\theta^e, \theta^d | \lambda)}{\partial \lambda}, \tag{13}$$

where we have $\partial \theta_{N_S}^e / \partial \theta_{N_P}^e$ to be the identity matrix due to the gradient approximation, $\partial \theta_{N_P}^e / \partial \ell_{\text{train}}^p(\theta^e, \theta^d | \lambda) = 1 / \frac{\partial \ell_{\text{train}}^p(\theta^e, \theta^d | \lambda)}{\partial \theta_{N_P}^e}$ which can be simply achieved at the end of the self-supervised training, and $\partial \ell_{\text{train}}^p(\theta^e, \theta^d | \lambda) / \partial \lambda$ can be obtained via back-propagation. The overall first-order approximation algorithm is given in Algorithm 1. After the joint training, there will be a final round of finetuning on the target task alone. Experimentally, we find this stage is useful when the target task has very few examples, and its contribution decreases as more training examples become available.

## 4 Experiments

We evaluate our proposed algorithm, referred to as **AutoSelect**, using the openly accessible MIMIC-III dataset [17] which contains over 38,000 adult patients admitted to the intensive care unit. We select a set of 96 common clinical measurements, which constitutes the set of candidate auxiliary tasks used for trajectory forecast. All values were normalized using z-score, and missing values were imputed by carrying forward the last observation. Yet, the model is always trained only using the true values as the targets instead of the imputed values.

We consider three primary supervised learning tasks defined using the criteria in Table 1. The prediction uses data from the first 48 hours of the ICU admission, and the label is positive if the criteria are fulfilled within the next 48 hour window (i.e. $48 - 96$ hours post admission). Moreover, the event sequence of each patient is also restricted

| Task | Definition |
|---|---|
| Mortality | Patient expired |
| Low Blood Pressure (BP) | Mean blood pressure $\leq$ 65mmhg |
| Kidney Dysfunction (KD) | Creatinine $\geq$ 2mg/dl |

Table 1: Task definitions.

to a window of 48 hours in the past, so that the bias towards longer stays can be alleviated. For simplicity, the latter two organ failure tasks are defined in a lightweight manner following the SOFA score criteria [18]. We report the details of the inclusion and exclusion criteria, the cohort and feature statistics, data preprocessing methods and results on additional tasks in the Appendix.

### 4.1 Baselines and Experiment Setting

**Supervised Learning.** We train a single baseline model with exactly the same architecture as the model used for the primary tasks of Table 1 in AutoSelect. Given that these primary tasks often have low resources, we expect supervised learning to have low predictive performance in general.

| Task | Data | Supervised | Pretrain (All) | CoTrain | AutoSelect |
|------|------|-----------|---------------|---------|-----------|
| Mortality | 1% | 0.738 (0.017) | 0.809 (0.010) | 0.725 (0.014) | **0.833** (0.017) |
| | 10% | 0.853 (0.016) | 0.853 (0.013) | 0.854 (0.014) | **0.882** (0.012) |
| | 100% | 0.899 (0.008) | 0.899 (0.011) | 0.902 (0.009) | **0.909** (0.008) |
| BP | 1% | 0.730 (0.022) | 0.778 (0.031) | 0.718 (0.041) | **0.838** (0.022) |
| | 10% | 0.754 (0.040) | 0.772 (0.028) | 0.724 (0.031) | **0.833** (0.018) |
| | 100% | 0.886 (0.026) | 0.881 (0.030) | 0.892 (0.018) | **0.899** (0.021) |
| KD | 1% | 0.745 (0.015) | 0.771 (0.021) | 0.748 (0.020) | **0.823** (0.018) |
| | 10% | 0.849 (0.015) | 0.828 (0.012) | 0.849 (0.012) | **0.862** (0.018) |
| | 100% | 0.901 (0.011) | 0.907 (0.007) | 0.899 (0.009) | **0.910** (0.011) |

Table 2: Predictive performance (AUC-ROC) of different competing methods for the three primary outcome prediction tasks under consideration with respect to different levels of data-scarcity.

**Pretraining (All).** This is the same pretraining-and-finetuning paradigm as in the work of [19]. We first learn the patient representation by pretraining using all the 96 self-supervised trajectory forecast tasks, and then finetune the model on the target tasks in Table 1.

**CoTrain via Multitask Learning.** This is a simple widely used multitask learning setup. We first cotrain the target task with all the auxiliary tasks, and use a task weight hyperparameter to balance the losses between these two groups of tasks. We set the weight of the target loss to be 10 tuned by the validation set performance to make the losses in the same scale, and the auxiliary task has a weight of 1. After the co-training stage, we finetune the model using only the data of the primary task.

**Experimental Setup.** The sequence-to-sequence architecture of the trajectory forecast task uses an LSTM for both encoder and decoder, where the hidden state has a dimension of 70. For the primary clinical outcome prediction task, the decoder is replaced by a simple 1-layer MLP as the classification head. All the baselines and our method use the same architecture. This does not prevent our method from using more advanced architectures. All models were implemented* in TensorFlow [20].

For the direct supervised learning baseline, we use early stopping with the validation set to avoid overfitting. For all the experiments of the other approaches, we run approximately 5,000 steps during the pretraining stage, followed by 5 epochs for finetuning. For AutoSelect, these 5,000 steps are further divided between inner loop steps and outer meta-learning steps, so that the total number of training steps is consistent with other pretrained methods for fair comparison. The learning rates of all training loops were tuned and are 0.001 for supervised learning, 0.005 for self-supervised learning, 0.01 for $\lambda$ hyper-gradient update. Detailed hyperparameter tuning process is reported in the Appendix. Finally, we use 10-fold cross validation and estimate the standard error of the mean. For each fold, we split the dataset into train/validation/test according to $80\%/10\%/10\%$ based on the hash value of the patient ID, and AUC-ROC is used as the evaluation metric by default with standard error reported in the parentheses next to it.

## 4.2 Performance Comparison of Clinical Outcome Prediction

Table 2 shows the results of gradually adding more training data by taking 1%, 10% and 100% from the original train dataset. We first observe that the performance of all methods in all tasks increases as more training data from the primary task are used. In the low resource regime, 'Pretrain (All)' has better performance than naive supervised learning, which is expected since it can transfer knowledge by learning from the auxiliary trajectory forecast tasks. Second, we also observe that the 'CoTrain' baseline has a hard time to balance all the 96 self-supervised trajectory forecasts and the supervised outcome prediction task even if it has an additional finetuning process. More sophisticated mixing ratios are thus needed to reconcile the different training speed of each task. Finally, we further compare AutoSelect to the two-stage pipeline approach [21] in the extreme case of using 1% of the data where it achieves 0.751(0.013), 0.720(0.022), 0.760(0.025) on the task of Mortality, BP and KD, respectively. By comparison, AutoSelect learns to adaptively tune the weight of each auxiliary task guided by the validation error from the primary task during pretraining, and thus is able to outperform these baselines by a significantly large margin.

| Table 3: Predictive performance of AutoSelect in selected tasks. | | | | | |
|---|---|---|---|---|---|
| Task | Data | AutoSelect | Pretrain (Top) | Pretrain (Down) | Pretrain (All) |
| BP | 1% | 0.838 (0.022) | 0.812 (0.014) | 0.788 (0.019) | 0.778 (0.031) |
| | 10% | 0.833 (0.018) | 0.824 (0.021) | 0.781 (0.027) | 0.772 (0.028) |
| KD | 1% | 0.823 (0.018) | 0.805 (0.016) | 0.749 (0.028) | 0.771 (0.021) |
| | 10% | 0.862 (0.018) | 0.855 (0.021) | 0.825 (0.021) | 0.828 (0.012) |
| Mortality | 1% | 0.833 (0.017) | 0.810 (0.013) | 0.772 (0.019) | 0.809 (0.010) |
| | 10% | 0.882 (0.012) | 0.850 (0.011) | 0.823 (0.014) | 0.853 (0.013) |

Table 4: Generalization of AutoSelect

| Data | Mortality$\rightarrow$ BP | Mortality$\rightarrow$ KD |
|---|---|---|
| 1% | 0.842 (0.019) | 0.833 (0.017) |
| 10% | 0.847 (0.020) | 0.869 (0.019) |

| Data | BP$\rightarrow$ Mortality | KD$\rightarrow$ Mortality |
|---|---|---|
| 1% | 0.812 (0.020) | 0.809 (0.018) |
| 10% | 0.871 (0.012) | 0.867 (0.013) |

## 4.3 Ablation Study

**What tasks are selected?** We now examine the pretraining tasks that were assigned with higher weights in the meta learning process shown in Figure 2b. The following features were consistently ranked within the top 20 across different training data splits for mortality prediction: invasive and non-invasive blood pressures, heart rate, anion gap, respiratory rate (Full list is available in the Appendix). These represent a mixture of common vital signs and laboratory values that are clinically sensible correlates for mortality. Indeed, there is significant overlap with the input features for classical risk scores that have been validated as mortality predictors in intensive care (e.g. APACHE II [22]). The top features for the other two supervised tasks, kidney dysfunction and low blood pressure, are detailed in the Appendix. Notably, the top features for low blood pressure include all available blood pressure recordings; however in the kidney dysfunction task, creatinine, which is the laboratory value on which the outcome is defined, does not appear in this top list. Our hypothesis is that creatinine is measured sparsely - typically once every 24 hours - thus providing a weak signal over the 48 hour window of the self-supervising trajectory forecast task.

**How good are the selected tasks?** To further validate the quality of top selected tasks and evaluate the impact of these features as pretraining tasks on the supervised outcome, we have conducted two ablation studies. In the first, we pretrain the encoder using the top selected auxiliary tasks only, referred to as 'Pretrain (Top)'. In the second, we instead pretrain the model with the remaining auxiliary tasks excluding the top ones, referred to as 'Pretrain (Down)'. The hypothesis is that the top selected tasks already capture the necessary knowledge needed to optimize the performance of the target task, while the remaining ones are less important. We report the results of these two studies in Table 3. It shows that 'Pretrain (Top)' performs closer to AutoSelect and is consistently better than 'Pretrain (Down)' and 'Pretrain (Full)', suggesting that the top selected tasks are able to transfer the most useful information to the target task. Meanwhile,we also observe that 'Pretrain (Down)' has similar performance to 'Pretrain (Full)' showing that the useful signals are indeed overshadowed in the learned representation with the full set of tasks.

**How does the learning occur?** Figure 2a presents the training dynamics of AutoSelect for the mortality task as an example. During the pretraining stage, its performance measured by AUC-ROC in the validation set keeps improving and then jumps even higher when the finetuning stage starts at step 5,500 when the validation performance of the auxiliary tasks reaches the peak. The performance of mortality prediction then quickly decreases due to overfitting to its small finetuning data. The blue curve represents the learning process of 'Pretrain (All)'. Because the mortality task was not involved in the pretraining stage, it only starts from the beginning of finetuning. The yellow curve is the process of 'CoTrain', where the validation performance of the mortality task first jumps and then decreases during the pretraining period. This is caused by the learning speed difference among all the tasks where the training of the small primary task starts to overfit while that of the other auxiliary tasks still improves. Finally, we study the impact of different training steps of the nested learning loops of AutoSelect. By fixing the total number of iterations around 5,000, in Figure 2c, we explore different configurations by varying the number of self-supervised training iterations ($N_P$) at the inner loop from 1,000 steps to 10 steps where (1,000/5) means 1,000 inner loop iterations and 5 outer loop iterations. In addition, the inner supervised training loop ($N_S$) is configured to take $1/10$ of the self-supervised iterations ($N_P$). We observe AutoSelect is generally robust across different configurations, and sweet points seem to be around (100/50) and (50/100).

**How does AutoSelect generalize?** Finally, we look at how well the learned weights of the auxiliary tasks guided by a given target task are able to generalize to unseen new tasks. We first use the mortality prediction task as the given primary task to guide the selection of the auxiliary trajectory forecast tasks. Then, we treat BP and KD as two new tasks, and directly finetune the model using 1% and 10% of the train data respectively. The reason is that, from a clinical perspective, mortality is a

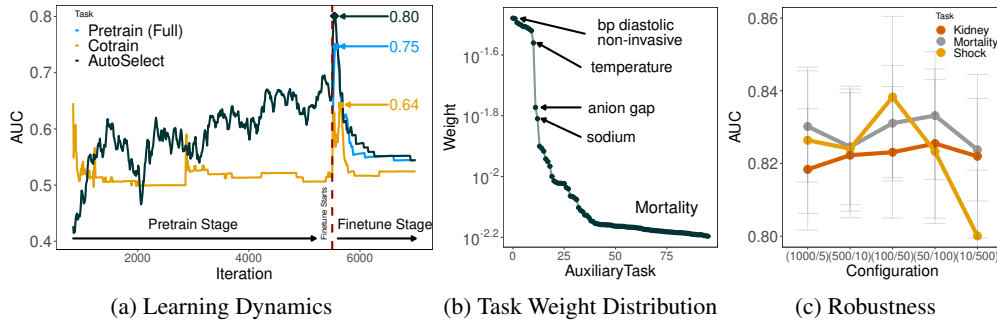

| (a) Learning Dynamics | (b) Task Weight Distribution | (c) Robustness |

Figure 2: (a) Pretraining and finetuning curves of competing methods. AUC-ROC on the validation set is reported. (b) The learned weights of the 96 auxiliary tasks for mortality prediction as the primary task. (c) Predictive performance with respect to different meta-learning processes of AutoSelect.

more general endpoint than specific dysfunctions. The hypothesis is that representations learned on the more general endpoint of mortality will be useful for more specific tasks. The prediction results are given in the top row of Table 3. Despite training only on the mortality task, the learned weights of the auxiliary tasks are able to improve the performance of BP and KD compared to the 'Pretrain (All)' baseline. Next, we use BP and KD to guide the selection learning separately, and then test on the mortality task at the bottom of Table 3. Since the specific dysfunction does not always imply an endpoint of mortality, the predictive performance on mortality is slightly worse than the respective results of AutoSelect.

## 5   Related Work

**Multi-Task Learning and Task selection**. Recent research on multitask learning [23] aims at either generalizing from a source multitask learning domain to a target multitask domain [24], or learning a good trade-off among different tasks [25]. There have been approaches to the task scheduling problem via a separate two-stage pipeline [21, 26], and Requeima et al. [27] learns to adjust model architectures to automatically adapt to new tasks. Doersch and Zisserman [28] combines multiple self-supervised tasks to train useful visual representations. Being complementary, our method learns to automatically select a set of auxiliary tasks, each of which is a self-supervised time-series learning problem, so that pretraining on these tasks can lead to a better representation for a target supervised learning task in an end-to-end differentiable framework.

**Meta learning**. Meta learning[29–32] seeks to acquire an initialization optimized for a set of tasks from the same distribution [33–35, 32]. One challenge of meta learning frameworks is that they rely on manually-defined training tasks, and hand-crafting these tasks can be time-consuming. The work of [33] presents unsupervised methods for inducing an adaptive meta-training task distribution and the work of [34] automatically constructs tasks from unlabeled data. We address this challenge via automatic selection over a large number of self-supervised tasks. Our method is similar in spirit to the work of [35] in that both direct the meta-learning process using a supervised target task, but we differ in that [35] meta-learns an unsupervised learning rule, while our work meta-learns a self-supervised task weight distribution.

**Patient Representation Learning**. The self-supervised task in our work is related to recent progress on state representation learning especially via patient trajectories [7–10]. The objectives in these works are often reconstruction errors in the entire observation space, and thus are not incentivized to capture latent factors that are useful for downstream tasks. To address this issue, the work of [36, 37] learn state representations by predicting the future in latent space with a probabilistic contrastive loss, while our work directs the representation learning by reducing error on a downstream target task.

## 6   Conclusion

We demonstrate how to leverage trajectory forecasts over clinical observations as self-supervised pretraining tasks to improve the quality of clinical outcome predictions. We present an efficient algorithm for automatic task selection and primary task training within a nested-loop meta-learning process. Experiments on a real clinical dataset show that our architecture achieves superior predictive performance, in particular in low-data scenarios when the primary task has very few examples.

# 7    Broader Impact

This work presents a method for efficiently learning patient representations using EMR data. Although this is demonstrated with a subset of the full raw EMR, and for only a handful of clinical outcomes in intensive care patients, it is a proof-of-concept that may be useful for a range of other predictive modeling using various types of longitudinal health data. The impact may be greatest in low-data scenarios - e.g. clinical use-cases where labeling is very challenging or where there are few eligible patients in the EMR. The code for this method will be made available to the research community on GitHub.

There are numerous ethical considerations associated with any EMR modeling, which have been discussed in the literature [38, 39]. Issues include numerous biases in the observational EMR data, e.g. on the basis of gender, ethnicity or socioeconomic status, which can propagate into predictive models. These fairness considerations also apply to representation learning architectures as presented here.

Finally, if this method were to be brought forward to real world deployment in conjunction with a decision support tool, it would have to be subject to appropriate clinical safety review and trials across different populations, with consideration given to issues such as drift and robustness.

## Footnotes

*https://github.com/google-health/records-research/meta-learn-forecast-task

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
