[Supplementary Material]



# 8 Appendix

## 8.1 Full Algorithm

Based on Equation 11 to 13, the gradient of $\lambda$ (Equation 9) can be solely determined by the signals of both $\alpha$ and $\beta$. The full algorithm is given in Algorithm 2.

---

**Algorithm 2:** Full Algorithm for Automatic Task Selection

---

Randomly initialize $\theta_0^e, \theta_0^d, \theta_{N_P}^c$ and $\lambda$;

**for** $k = 1, 2, ...$ **do**

    **for** $i \in [1, N_P]$ **do**                          ▷ `Self-supervised learning loop`

        $\theta_i^e = \Psi_i^e \left(\theta_{i-1}^e, \theta_{i-1}^d, \lambda_k\right), \theta_i^d = \Psi_i^d \left(\theta_{i-1}^e, \theta_{i-1}^d, \lambda_k\right)$ ;     ▷ `Gradient descent`

        $\mathbf{H}_i^e \leftarrow \nabla_{\theta_i^e} \Psi_{i+1}^e(\theta_i^e, \theta_i^d, \lambda)$ ;             ▷ `Record Hessian Matrix`

        $\mathbf{J}_i^e \leftarrow \nabla_{\lambda} \Psi_i^e(\theta_{i-1}^e, \theta_{i-1}^d, \lambda)$ ;         ▷ `Record Jacobian for λ over Gradient`

    **for** $i \in [N_P + 1, N_P + N_S]$ **do**            ▷ `Supervised learning loop`

        $\theta_i^e = \Phi_i^e \left(\theta_{i-1}^e, \theta_{i-1}^c\right), \theta_i^c = \Phi_i^c \left(\theta_{i-1}^e, \theta_{i-1}^c\right)$;        ▷ `Gradient descent`

        $\mathbf{H}_i^e \leftarrow \nabla_{\theta_i^e} \Phi_{i+1}^e(\theta_i^e, \theta_i^c)$ ;             ▷ `Record Hessian Matrix`

    $\beta_{N_S} \leftarrow \nabla_{\theta_{N_S}^e} \ell_{\text{val}}^c$ ;                  ▷ `Initialize multiplier`

    **for** $i \in [N_P + N_S - 1, N_P + 1]$ **do**        ▷ `Reverse steps of supervised loop`

        $\beta_i \leftarrow \beta_{i+1} \mathbf{H}_i^e$ ;                 ▷ `Update multiplier`

    $\alpha_{N_P} = \beta_{N_P+1} \mathbf{H}_{N_P}^e$ ;         ▷ `Transfer Supervised to Self-supervised`

    $g \leftarrow 0$ ;                           ▷ `Initialize hyper-gradient`

    **for** $i \in [N_P, 1]$ **do**          ▷ `Reverse steps of self-supervised loop`

        $g_\lambda \leftarrow g_\lambda + \alpha_{i+1} \mathbf{J}_{i+1}^e$ ;           ▷ `Update hyper-gradient`

        $\alpha_i \leftarrow \alpha_{i+1} \mathbf{H}_i^e$ ;                 ▷ `Update multiplier`

    $\lambda_i = \lambda_{i-1} - \epsilon \cdot g_\lambda$ ;              ▷ `Hyper-gradient descent`

**return** $\theta_{N_P}^e, \lambda$

---

Note that the gradient descent steps follow the following equations:

$$\Psi_i^e(\theta_{i-1}^e, \theta_{i-1}^d, \lambda) = \theta_{i-1}^e - \eta^p \nabla_{\theta^e} \ell_{\text{train}}^p(\theta^e, \theta^d, \lambda) \tag{15}$$

$$\Phi_i^e(\theta_{i-1}^e, \theta_{i-1}^c) = \theta_{i-1}^e - \eta^c \nabla_{\theta^e} \ell_{\text{train}}^c(\theta^e, \theta^c) \tag{16}$$

where $\eta^p$ and $\eta^c$ are the learning rates for the self-supervised training and supervised training respectively. As thus, $\nabla_\lambda \Psi_i^e(\theta_{i-1}^e, \theta_{i-1}^d, \lambda)$, $\nabla_{\theta_i^e} \Psi_{i+1}^e(\theta_i^e, \theta_i^d, \lambda)$ and $\nabla_{\theta_i^e} \Phi_{i+1}^e(\theta_i^e, \theta_i^c)$ include the Jacobian and Hessian matrix of the gradient update operation $\Psi_i^e$ and $\Phi_{i+1}^e$ and are cached during the learning loops. Also to simplify notations, we overloaded $N_S$ and use $\beta_{N_S}$ and $\nabla_{\theta_{N_S}^e}$ to represent $\beta_{N_S+N_P}$ and $\nabla_{\theta_{N_S+N_P}^e}$ in the algorithm.

## 8.2 Dataset and Data Preprocessing

We evaluate our proposed algorithm using the open access MIMIC-III dataset [17] which contains patients admitted to the intensive care unit of Beth Israel Deaconess Medical Center between 2001 and 2012.

Table 4: Task definition and statistics.

| Task | Definition | Positive | Negative | Excluded |
|------|-----------|----------|----------|----------|
| Mortality | patient expired | 597 | 36,773 | 1,115 |
| Shock | mean blood pressure $\leq 65$ | 3,418 | 15,159 | 19,908 |
| Kidney Failure | creatinine $\geq$ 2mg/dl | 526 | 32,055 | 5,904 |
| Liver Failure | bilirubin $\geq$ 2mg/dl | 387 | 34,856 | 3,242 |

There are 38,485 adult inpatient encounters included in the study. Supervised tasks were triggered at 48 hrs after ICU admission with a lookahead horizon of 48 hours (i.e. 48-96 hours post admission) for the following endpoints: inpatient mortality, renal and liver failure, and circulatory shock. For this proof of concept work, the latter three organ failure endpoints were defined in a lightweight manner

based on SOFA score criteria [18], as described in Table 4, rather than using more detailed definitions taking into account baseline values.

The positive (negative) columns are the number of encounters where the corresponding events are observed (unobserved) after the prediction time (i.e., 48 hrs after ICU admission) and within the prediction window (i.e. 48-96 hours post admission). The excluded column counts the encounters where the corresponding event occurs before the prediction time. We only include the encounters greater than 48 hours in length and where the outcome of interest did not occur within the first 48 hrs.

We select the 96 most frequently occurring lab measurements, vital signs and interventions/equipment settings as features to define the trajectory forecast tasks, as listed in Table 9. As different coding systems are used in MIMIC-III, we harmonize the medical codes corresponding to the same lab/vital measurement as a single feature. In addition, we standardize the units when a medical code is used with multiple units or without a unit. The data preprocessing steps are summarized as follows:

1. **Code harmonization**. This is a manual process based on input from clinical experts. In this step, the medical codes corresponding to the same measurements from different coding systems, including LOINC and MIMIC specific coding, are harmonized into the same entity. For example, serum creatinine is associated with the following MIMIC-III specific codes: 220615, 50912, 1525, 3750, 791, and LOINC code 2160-0.

2. **Unit conversion**. This is an automated process with manual review. In MIMIC-III, a medical code may be stored in multiple units and sometimes the unit might be missing. To determine whether these two entries correspond to the same measurement concept, we derive its value range and mean under different units and test whether they are similar to each other. The final results are reviewed manually.

3. **Outlier removal**. We derive the distribution of each measurement code after harmonization and unit conversion. We remove the outliers, defined as $< 0.1\times$ the value at the 1st percentile or $> 10\times$ the value at the 99th percentile.

4. **Value normalization**. We collect the mean and standard deviation over the cleaned dataset for each harmonized code and compute its z-score as feature value.

5. **Time bucketing**. Each timestep in the experiment corresponds to 1 hour. If multiple measurements exist within this 1 hour for the same clinical feature, we take its average value.

In Table 9, the first column is the name and unit of the feature after harmonization. The second column includes all the names being used for this feature in the MIMIC-III dataset. The corresponding MIMIC-specific and LOINC codes are listed in the third and fourth columns. The 'Units' column lists all the original units associated with this feature. We use 'null' to denote the case where unit is left empty in the dataset. Finally, the last column lists the number of appearances of this feature in the dataset.

Table 5: Hyperparameter Selection.

| Hyperparameter | Values/Range Considered | Selection Criteria | Value Selected |
|---|---|---|---|
| Learning rate for supervised learning | {0.0005, 0.001, 0.05, 0.01} | Validation AUC for supervised training and fine-tune after pretraining | 0.001 |
| Learning rate for self-supervised learning | {0.0005, 0.001, 0.05, 0.01} | Validation MSE in pretraining | 0.005 |
| Learning rate for $\lambda$ update | {0.001, 0.01, 0.1} | Validation AUC for fine-tune after pretraining | 0.01 |
| State size for LSTM | Range [20, 100] with increment of 10 | Validation MSE in pretraining and validation AUC for supervised training | 70 |

| Task | Data | Supervised | Pretrain (All) | CoTrain | AutoSelect |
|------|------|-----------|----------------|---------|------------|
| Mortality | 1% | 0.738 (0.017) | 0.809 (0.010) | 0.725 (0.014) | **0.833** (0.017) |
| | 10% | 0.853 (0.016) | 0.853 (0.013) | 0.854 (0.014) | **0.882** (0.012) |
| | 100% | 0.899 (0.008) | 0.899 (0.011) | 0.902 (0.009) | **0.909** (0.008) |
| Shock | 1% | 0.730 (0.022) | 0.778 (0.031) | 0.718 (0.041) | **0.838** (0.022) |
| | 10% | 0.754 (0.040) | 0.772 (0.028) | 0.724 (0.031) | **0.833** (0.018) |
| | 100% | 0.886 (0.026) | 0.881 (0.030) | 0.892 (0.018) | **0.899** (0.021) |
| Kidney F. | 1% | 0.745 (0.015) | 0.771 (0.021) | 0.748 (0.020) | **0.823** (0.018) |
| | 10% | 0.849 (0.015) | 0.828 (0.012) | 0.849 (0.012) | **0.862** (0.018) |
| | 100% | 0.901 (0.011) | 0.907 (0.007) | 0.899 (0.009) | **0.910** (0.011) |
| Liver F. | 1% | 0.721 (0.017) | 0.780 (0.010) | 0.707 (0.027) | **0.822** (0.013) |
| | 10% | 0.863 (0.015) | 0.832 (0.009) | 0.832 (0.009) | **0.866** (0.016) |
| | 100% | 0.896 (0.011) | 0.896 (0.009) | 0.896 (0.008) | **0.908** (0.007) |

Table 6: Predictive performance (AUC-ROC) of different competing methods for the four primary outcome prediction tasks under consideration with respect to different levels of data-scarcity.

| Task | Data | Supervised | Pretrain (All) | CoTrain | AutoSelect |
|------|------|-----------|----------------|---------|------------|
| Mortality | 1% | 0.047 (0.004) | 0.073 (0.007) | 0.056 (0.007) | **0.097** (0.009) |
| | 10% | 0.127 (0.011) | 0.154 (0.015) | 0.135 (0.009) | **0.168** (0.024) |
| | 100% | 0.267 (0.028) | 0.268 (0.019) | 0.262 (0.028) | **0.291** (0.033) |
| Shock | 1% | 0.035 (0.005) | 0.052 (0.015 | 0.054 (0.019) | **0.080** (0.019) |
| | 10% | 0.086 (0.025) | 0.096 (0.029) | 0.082 (0.021) | **0.109** (0.031) |
| | 100% | 0.109 (0.031) | 0.206 (0.052) | 0.203 (0.057) | **0.206** (0.050) |
| Kidney F. | 1% | 0.047 (0.002) | 0.074 (0.013) | 0.075 (0.011) | **0.105** (0.009) |
| | 10% | 0.126 (0.014) | 0.113 (0.013) | 0.115 (0.014) | **0.144** (0.014) |
| | 100% | 0.204 (0.017) | 0.223 (0.012) | 0.226 (0.016) | **0.249** (0.017) |
| Liver F. | 1% | 0.040 (0.004) | 0.076 (0.016) | 0.046 (0.008) | **0.085** (0.011) |
| | 10% | 0.120 (0.015) | 0.107 (0.011) | 0.122 (0.015) | **0.156** (0.021) |
| | 100% | 0.229 (0.018) | 0.184 (0.012) | 0.226 (0.015) | **0.257** (0.015) |

Table 7: Predictive performance (AUC-PR) of different competing methods for the four primary outcome prediction tasks under consideration with respect to different levels of data-scarcity.

## 8.3 Hyperparameters and Selections

The learning rates of all training loops were tuned jointly with the state size of LSTM via a grid search. Table 5 shows the list of values considered, the criteria we use in choosing the hyperparameter and the final value selection.

## 8.4 Additional Experiment Results

Table 6 provides predictive performance (AUC-ROC) over the additional Liver Failure prediction task along with the other three tasks. We also report the AUC-PR (i.e., Average Precision (AP)) of different competing methods for the four primary outcome prediction tasks under consideration in Table 7. As we could see, AutoSelect outperforms both 'Pretrain (All)' and 'CoTrain' by a significantly large margin in low-data scenarios for all four tasks with respect to both metrics.

## 8.5 Top Tasks

The features that are associated with the top trajectory forecast auxiliary tasks for each primary task are listed in Table 8.

Table 8: Top Trajectory Forecast Tasks.

| Task | Top Features |
|------|--------------|
| Mortality | Invasive/Non-invasive blood pressure (diastolic, mean, systolic), anion gap, heart rate, respiratory rate, minute ventilation obs, urine output foley, temperature, o2 saturation p, o2 flow, magnesium, sodium, glucose poc, calcium, glucose, hemoglobin, paw, arterial base excess, mean chc, arterial bicarbonate, base excess, rbc. |
| Shock | Invasive/Non-invasive blood pressure (diastolic, mean, systolic), heart rate, respiratory rate, urine output foley, temperature, o2 saturation p, o2 flow, magnesium, sodium, glucose, hemoglobin, paw. |
| Kidney Failure | Invasive/Non-invasive blood pressure (diastolic, mean, systolic), heart rate, respiratory rate, urine output foley, temperature, o2 saturation p, magnesium, sodium, glucose. |
| Liver Failure | Invasive/Non-invasive blood pressure (diastolic, mean, systolic), anion gap, heart rate, respiratory rate, minute ventilation obs, urine output foley, temperature, o2 saturation p, magnesium, sodium, glucose poc, calcium, glucose, hemoglobin, mean chc. |

Table 9: List of input features used in the model.

| Harmonized Name | Display Names | MIMIC code | LOINC code | Units | Count |
|---|---|---|---|---|---|
| Access pressure @mmHg | Access Pressure, Access mmHg | 224149, 29 | | mmHg | 11490 |
| Albumin @G_PER_DL | Albumin, Albumin (3.9-4.8), Albumin (>3.2) | 1521, 227456, 3727, 50862, 772 | 1751-7 | G_PER_DL, null | 161658 |
| Alt @IU_PER_L | ALT, Alanine Aminotransferase (ALT) | 220644, 50861, 769 | 1742-6 | IU_PER_L, null | 244169 |
| Anion gap @MEQ_PER_L | Anion gap, Anion Gap (8-20), Anion Gap | 227073, 3732, 50868 | 1863-0 | MEQ_PER_L, null | 869575 |
| Ap @IU_PER_L | Alkaline Phosphate, Alkaline Phosphatase, Alk. Phosphate | 225612, 3728, 50863, 773 | 6768-6 | null, IU_PER_L | 232122 |
| Arterial base excess @MEQ_PER_L | Arterial Base Excess | 224828, 776 | | null, MEQ_PER_L | 271605 |
| Arterial bicarbonate @MEQ_PER_L | TCO2 (calc) Arterial, Arterial CO2(Calc) | 225698, 777 | | MEQ_PER_L | 406734 |
| Arterial pco2 @mmHg | Arterial CO2 Pressure, Arterial PaCO2 | 220235, 778 | | mmHg | 406498 |
| Arterial ph @PH | Art.pH, PH (Arterial), pH (Art), Arterial pH | 1126, 223830, 4753, 780 | | PH, null | 430827 |
| Arterial po2 @mmHg | Arterial O2 pressure, Arterial PaO2 | 220224, 779 | | mmHg | 406341 |
| Ast @IU_PER_L | AST, Asparate Aminotransferase (AST) | 220587, 50878, 770 | 1920-8 | null, IU_PER_L | 244187 |
| Base excess @MEQ_PER_L | Base Excess | 50802 | 11555-0 | null, MEQ_PER_L | 329827 |
| Basophils @PERCENT | Basophils | 51146 | 704-7 | null, PERCENT | 172039 |
| Blood flow @ML_PER_MIN | Blood Flow (ml/min), Blood Flow ml/min | 224144, 79 | | ML_PER_MIN | 116654 |
| Bp diastolic invasive @mmHg | Arterial Blood Pressure diastolic, ART BP Diastolic, Arterial BP [Diastolic], Arterial BP 2 [Diastolic] | 220051, 225310, 8368, 8555 | | mmHg | 3292274 |
| Bp diastolic non invasive @mmHg | Non Invasive Blood Pressure diastolic, Manual Blood Pressure Diastolic Left, Manual Blood Pressure Diastolic Right, Manual BP [Diastolic], NBP [Diastolic] | 220180, 224643, 227242, 8440, 8441 | | mmHg | 2863944 |
| Bp map invasive @mmHg | Arterial Blood Pressure mean, ART BP mean, Arterial BP Mean, Arterial BP Mean 2, Arterial Mean 3 | 220052, 225312, 52, 6702, 6927 | | mmHg | 3280702 |

Table 9: List of input features used in the model.

| Harmonized Name | Display Names | MIMIC code | LOINC code | Units | Count |
|---|---|---|---|---|---|
| Bp mean non invasive @mmHg | Non Invasive Blood Pressure mean, Manual BP Mean(calc), NBP Mean | 220181, 443, 456 | | mmHg | 2843856 |
| Bp systolic invasive @mmHg | Arterial Blood Pressure systolic, ART BP Systolic, Arterial BP [Systolic], Arterial BP 2 [Systolic] | 220050, 225309, 51, 6701 | | mmHg | 3293052 |
| Bp systolic non invasive @mmHg | Non Invasive Blood Pressure systolic, Manual Blood Pressure Systolic Left, Manual Blood Pressure Systolic Right, Manual BP [Systolic], NBP [Systolic] | 220179, 224167, 227243, 442, 455 | | mmHg | 2865402 |
| Bun @MG_PER_DL | BUN, BUN (6-20), Urea Nitrogen, BUN (6-20) | 1162, 225624, 3737, 51006, 781 | 3094-0 | MG_PER_DL, null | 893969 |
| Calcium @ | Calcium, Calcium non-ionized, Calcium (8.8-10.8), Calcium, Total, Calcium (8.4-10.2) | 1522, 225625, 3746, 50893, 786 | 2000-8 | MEQ_PER_L, null | 323 |
| Calcium @MEQ_PER_L | Calcium, Calcium non-ionized, Calcium (8.8-10.8), Calcium, Total, Calcium (8.4-10.2) | 1522, 225625, 3746, 50893, 786 | 2000-8 | MEQ_PER_L, null | 681992 |
| Cardiac index @UNKNOWN_UOM | Cardiac Index, Cardiac Index (CI NICOM) | 116, 228368 | | UNKNOWN_UOM, L_PER_MIN_PER_M2 | 209483 |
| Cardiac output rate @L_PER_MIN | Cardiac Output (thermodilution), Cardiac Output (CCO), CO (Arterial), CO (PiCCO), Cardiac Output (CO NICOM), C.O. (fick), C.O.(thermodilution) | 220088, 224842, 227543, 228178, 228369, 89, 90 | | L_PER_MIN | 294655 |
| Chloride @MEQ_PER_L | Chloride, Chloride (serum), Chloride (whole blood), Chloride (100-112), Chloride, Whole Blood, Chloride (100-112) | 1523, 220602, 226536, 3747, 50806, 50902, 788 | 2069-3, 2075-0 | MEQ_PER_L, null | 959368 |
| Co2 @MEQ_PER_L | HCO3 (serum), Calculated Bicarbonate, Whole Blood, Bicarbonate, Carbon Dioxide, HCO3 | 227443, 50803, 50882, 787, 812 | 1959-6, 1963-8 | null, MEQ_PER_L | 891999 |
| Creatine kinase @IU_PER_L | CK (CPK), Creatine Kinase (CK), CPK | 225634, 50910, 784 | 2157-6 | IU_PER_L, null | 149871 |
| Creatinine @MG_PER_DL | Creatinine, Creatinine (0-0.7), Creatinine (0-1.3) | 1525, 220615, 3750, 50912, 791 | 2160-0 | MG_PER_DL, null | 899717 |
| Creatinine kinase mb @NG_PER_ML | CK-MB, Creatine Kinase, MB Isoenzyme, CPK/MB | 227445, 50911, 785 | 6773-6 | NG_PER_ML, null | 89737 |
| Cvp @mmHg | CVP, Central Venous Pressure | 113, 220074 | | mmHg, PERCENT | 1680374 |
| Eosinophils @PERCENT | Eosinophils | 51200 | 711-2 | null, PERCENT | 172042 |
| Exhaled minute ventilation low @ | High exhaled min vol, high exhaled min vol, HIGH EXHALED MIN VOL, High Exhaled min vol, high exhaled min.vol, High exhaled min.vol, high exhale MV, HIGH EXHALE MV, High exhaled MV, High exhale MV, high exhale mv, Low Exhaled Min Vol, High Exhaled Min vol | 1010, 1102, 1223, 1313, 1323, 1380, 1720, 1723, 1724, 2123, 2127, 434, 5744 | | L_PER_MIN, null | 186 |

Table 9: List of input features used in the model.

| Harmonized Name | Display Names | MIMIC code | LOINC code | Units | Count |
|---|---|---|---|---|---|
| Exhaled minute ventilation low @L_PER_MIN | High exhaled min vol, high exhaled min vol, HIGH EXHALED MIN VOL, High Exhaled min vol, high exhaled min.vol, High exhaled min.vol, high exhale MV, HIGH EXHALE MV, High exhaled MV, High exhale MV, high exhale mv, Low Exhaled Min Vol, High Exhaled Min vol | 1010, 1102, 1223, 1313, 1323, 1380, 1720, 1723, 1724, 2123, 2127, 434, 5744 | | L_PER_MIN, null | 386163 |
| Expiratory ratio @RATIO | Expiratory Ratio | 226871 | | RATIO | 207217 |
| Fio2 analyzed @TORR | FiO2 (Analyzed) | 189 | | TORR | 6914 |
| Glucose @MG_PER_DL | Glucose, Glucose (serum), Glucose (whole blood), Glucose (70-105) | 1529, 220621, 226537, 50809, 50931, 811 | 2339-0, 2345-7 | MG_PER_DL, null | 1172288 |
| Glucose poc @MG_PER_DL | Glucose finger stick, Fingerstick Glucose | 225664, 807 | | MG_PER_DL | 678665 |
| Heart rate @BPM | Heart Rate | 211, 220045 | | BPM | 7938853 |
| Hematocrit @PERCENT | Hematocrit (serum), Hematocrit (whole blood - calc), Hematocrit (35-51), Hematocrit, Calculated, Hematocrit | 220545, 226540, 3761, 50810, 51221, 813 | 20570-8, 4544-3 | null, PERCENT | 1094791 |
| Hemoglobin @G_PER_DL | Hemoglobin, HGB (10.8-15.8), Absolute Hemoglobin | 220228, 3759, 50811, 50855, 51222, 814 | 718-7 | G_PER_DL, null | 942483 |
| Inr @RATIO | INR, INR(PT), INR (2-4 ref. range) | 1530, 227467, 51237, 815 | 5895-7 | RATIO, null | 161338 |
| Inr @ | INR, INR(PT), INR (2-4 ref. range) | 1530, 227467, 51237, 815 | 5895-7 | RATIO, null | 371373 |
| Insp pressure @CM_H2O | High Insp. Pressure, low insp pressure, Low Insp. Pressure, Low insp pressure, low IP | 218, 3143, 436, 6864, 7094 | | CM_H2O | 384580 |
| Insp time @S | Insp. Time, Inspiratory Time, Insp.Time, Insp time | 1655, 2000, 224738, 3009, 6315 | | null, S | 364845 |
| Inspiratory ratio @RATIO | Inspiratory Ratio | 226873 | | RATIO | 206635 |
| Ionized calcium @MEQ_PER_L | Ionized Calcium, Free Calcium | 225667, 50808, 816 | 1994-3 | null, MEQ_PER_L | 303125 |
| Ketones urine @MG_PER_DL | Ketone | 51484 | 5797-6 | MG_PER_DL | 10617 |
| Lactate @MMOL_PER_L | Lactic Acid, Lactate, Lactic Acid(0.5-2.0) | 1531, 225668, 50813, 818 | 32693-4 | MMOL_PER_L, null | 233021 |
| Lymphocytes diff @PERCENT | Differential-Lymphs | 225641, 798 | | PERCENT | 41425 |
| Magnesium @MG_PER_DL | Magnesium, Magnesium (1.6-2.6) | 1532, 220635, 50960, 821 | 2601-3 | MG_PER_DL, null | 762825 |
| Mean ch @PG | MCH | 51248 | 785-6 | null, PG | 747387 |
| Mean chc @PERCENT | MCHC | 51249 | 786-4 | null, PERCENT | 747756 |
| Mean cv @FL | MCV | 51250 | 787-2 | null, FL | 747377 |
| Minute ventilation obs @L_PER_MIN | Minute Volume, Minute Volume(Obser) | 224687, 448, 450 | | L_PER_MIN | 851154 |
| Monocytes @PERCENT | Monocytes | 51254 | 742-7 | null, PERCENT | 172044 |
| Neutrophils urine @PERCENT | Neutrophils | 51256 | 761-7 | null, PERCENT | 170591 |

Table 9: List of input features used in the model.

| Harmonized Name | Display Names | MIMIC code | LOINC code | Units | Count |
|---|---|---|---|---|---|
| O2 flow @L_PER_MIN | O2 Flow, O2 Flow (additional cannula), O2 Flow (lpm), O2 Flow (lpm) 2 | 223834, 227287, 470, 471, 50815 | 3151-8 | L_PER_MIN | 622606 |
| O2 saturation @PERCENT | Arterial O2 Saturation, Oxygen Saturation, SaO2 | 220227, 50817, 834 | 20564-1 | PERCENT, null | 1828799 |
| O2 saturation p @PERCENT | O2 saturation pulseoxymetry, SpO2, SpO2-L | 220277, 646, 6719 | | PERCENT | 6086176 |
| P co2 @mmHg | pCO2 | 50818 | 11557-6 | null, mmHg | 490470 |
| P o2 @mmHg | pO2 | 50821 | 11556-8 | null, mmHg | 490481 |
| Paw @CM_H2O | MEAN AIRWAY PRESS, PAW, Paw High, Mean Airway Pressure, Mean PAW, Mean PAW [Meas] | 1672, 2229, 223873, 224697, 3502, 3503, 444 | | null, CM_H2O | 1101930 |
| Peep observed @CM_H2O | PEEP, MEASURED PEEP | 505, 6924 | | CM_H2O, null | 350192 |
| Ph @PH | PH, pH | 1673, 50820 | 11558-4 | null, PH | 530708 |
| Ph urine @PH | urine pH, urine ph, Urine pH, PH (dipstick), pH, urine PH | 1352, 1495, 1880, 220734, 51491, 6754 | 5803-2 | PH, null | 129352 |
| Phosphorous @MEQ_PER_L | Phosphorous, Phosphate, Phosphorous(2.7-4.5) | 1534, 225677, 50970, 827 | 2777-1 | null, MEQ_PER_L | 681128 |
| Plateau pressure @CM_H2O | Plateau Pressure | 224696, 543 | | CM_H2O | 242730 |
| Platelet @ | Platelet Count, Platelet (150-440), Platelets | 227457, 3789, 51265, 828 | 777-3 | K_PER_UL, null | 5313 |
| Platelet @K_PER_UL | Platelet Count, Platelet (150-440), Platelets | 227457, 3789, 51265, 828 | 777-3 | K_PER_UL, null | 865223 |
| Potassium @MEQ_PER_L | Potassium, Potassium (serum), Potassium (whole blood), Potassium (3.5-5.3), Potassium, Whole Blood, Potassium (3.5-5.3) | 1535, 227442, 227464, 3792, 50822, 50971, 829 | 2823-3, 6298-4 | MEQ_PER_L, null | 1181418 |
| Potassium urine @MEQ_PER_L | Potassium, Urine | 51097 | 2828-2 | null, MEQ_PER_L | 10850 |
| Protein urine @MG_PER_DL | Protein | 51492 | 5804-0 | null, MG_PER_DL | 36043 |
| Psv @UNKNOWN_UOM | PSV Level, Pressure Support, pressure support, PSV | 224701, 578, 7332, 7595 | | CM_H2O, null, UNKNOWN_UOM | 406535 |
| Pt @S | PT, Prothrombin time, Pro-Time, PT(11-13.5) | 1286, 227465, 3793, 51274, 824 | 5902-2 | S, null | 530740 |
| Ptt @S | PTT, Ptt, PTT(22-35) | 1533, 227466, 3796, 51275, 825 | 3173-2 | S, null | 539725 |
| Rbc @PER_UL | Red Blood C(3.6-6.2), RBC(3.6-6.2), Red Blood Cells, RBC | 3799, 4197, 51279, 833 | 789-8 | PER_UL, null | 748036 |
| Rdw @PERCENT | RDW | 51277 | 788-0 | null, PERCENT | 746239 |
| Replacement rate @ML_PER_H | Replacement Rate, Replace Rate ml/hr | 224153, 611 | | ML_PER_H | 118496 |
| Respiratory rate @BREATHS_PER_MIN | High Resp. Rate, Respiratory Rate, high rr, Resp Rate | 219, 220210, 3142, 3603, 618 | | BREATHS_PER_MIN | 7904015 |
| Respiratory rate spont @BREATHS_PER_MIN | Spont Resp rate, Spont RR, Respiratory Rate (spontaneous), Resp Rate (Spont), Spon RR (Mech.), Spont. Resp. Rate | 1884, 224422, 224689, 614, 651, 653 | | null, BPM, BREATHS_PER_MIN | 714134 |

Table 9: List of input features used in the model.

| Harmonized Name | Display Names | MIMIC code | LOINC code | Units | Count |
|---|---|---|---|---|---|
| Respiratory rate total @BPM | Respiratory Rate (Total), Resp Rate (Total) | 224690, 615 | | BREATHS_PER_MIN, BPM | 812342 |
| Rrt output @ML | Ultrafiltrate Output, Ultrafiltrate Ultrafiltrate, dialysis output, dialysis out, Dialysis out, DIALYSIS OUT, dialysis, DIALYSIS, Dialysis, PD dialysate out, Dialysis Out, dialysate out, Dialysis output, ULTRAFILTRATE, ultrafiltrate in, peritoneal dialysis, ultrafiltrate, ultrafiltrate out | 226457, 40286, 40425, 40426, 40507, 40613, 40624, 40690, 40745, 40789, 41374, 41623, 42536, 43703, 44349, 44843, 44890, 46622 | | ML, null | 123788 |
| Sodium @MEQ_PER_L | Sodium, Sodium (serum), Sodium (whole blood), Sodium (135-148), Sodium, Whole Blood, Sodium (135-148) | 1536, 220645, 226534, 3803, 50824, 50983, 837 | 2947-0, 2951-2 | MEQ_PER_L, null | 1000050 |
| Svo2 @PERCENT | SVO2, SvO2, Mixed Venous O2% Sat, svo2, Swan SVO2, SV02, svo2 RIJ | 2194, 223772, 225674, 2669, 664, 7361, 8186, 823, 838 | | PERCENT | 250574 |
| Tbili @MG_PER_DL | Total Bili, Total Bilirubin, Bilirubin, Total, Total Bili (0-1.5) | 1538, 225690, 50885, 848 | 1975-2 | MG_PER_DL, null | 263182 |
| Tco2 @MEQ_PER_L | TCO2 (21-30), Calculated Total CO2 | 3808, 50804 | 34728-6 | null, MEQ_PER_L | 490499 |
| Temperature @CEL | Temperature Fahrenheit, Temperature Celsius, Temp Axillary [F], Temp Rectal [F], Temperature C, Temperature F | 223761, 223762, 3652, 3654, 676, 678 | | CEL | 2203935 |
| Troponin t @NG_PER_ML | Troponin-T, Troponin T | 227429, 51003 | 6598-7 | NG_PER_ML | 79888 |
| Urine output @ML | Void, OR Urine, PACU Urine, Drainage Bag, OR Out OR Urine, OR Out PACU Urine, Urine Out Void, Urine Out Incontinent, PACU Out PACU Urine, Urine Out Other, ER URINE, TRUE URINE, Urine ., Dialysis indwelling, True Urine | 226560, 226627, 226631, 227701, 40061, 40065, 40069, 40085, 40288, 40405, 42001, 42507, 43175, 44286, 45927 | | ML, null | 252169 |
| Urine output foley @ML | Foley, Urine Out Foley | 226559, 40055 | | ML, null | 3093578 |
| Vt obs @ML_PER_BREATH | Tidal Volume (observed), tidal volumes, tidal vol, tidal volume, Tidal Volume, Tidal Volume (Obser) | 224685, 2400, 2408, 2534, 681, 682 | | ML_PER_BREATH | 701861 |

Table 9: List of input features used in the model.

| Harmonized Name | Display Names | MIMIC code | LOINC code | Units | Count |
|---|---|---|---|---|---|
| Vt spont @ML_PER_BREATH | Spont Vt, Tidal Volume (spontaneous), svt, Spontaneous VT, spontaneous VT, spont tidal volumes, spont Tidal volumes, spont Vt's, Spon. Vt (L) (Mech.), Spont. Tidal Volume, Tidal Volume (Spont) | 224421, 224686, 2553, 2566, 3004, 3050, 3083, 3086, 652, 654, 684 | | ML_PER_BREATH, null | 494654 |
| Wbc count @K_PER_UL | WBC (4-11,000), WBC, WBC 4.0-11.0, White Blood Cells, WBC (4-11,000) | 1127, 1542, 220546, 4200, 51301, 861 | 804-5 | K_PER_UL, null | 842129 |
| Weight @KG | Daily Weight | 224639, 763 | | KG | 93386 |