[Reviews · NeurIPS 2020]

Review 1

Summary and Contributions: This paper proposes a method to automatically select auxiliary tasks that the representations pretrained from which are most useful and informative for a given primary target task. Specifically, they develop an efficient algorithm that uses automatic auxiliary task selection within a nested-loop meta-learning process. They apply this method to the target task of clinical outcome prediction, with the auxiliary tasks of patient trajectories forecasting. Experiments on real-world EMR data shows the effectiveness of the method.

Strengths: 1. The method aims to solve an important problem in transfer learning where traditional strategies are inefficient and suboptimal, by proposing an automatic auxiliary task selection method. The idea is novel and has practical meaning for related research. 2. The paper is well written with thorough analysis. 3. It is interesting to see the generalization experiment of AutoSelect, which may shed some light on the relationship between generally and specific clinical outcome prediction.

Weaknesses: 1. AUC-ROC is the only metric in all the experiments, which could make the results less convincing. Other metrics, such as the recall rate for positive samples (sensitivity), are also important for clinical outcome prediction and should be used for evaluation. 2. Table 2 only shows comparison with three simple strategies, i.e., supervised, pretrain(all), and cotrain, and lacks proper comparison with other task selection baselines, such as those mentioned in related work. It would be more convincing if such comparison can be conducted.

Correctness: It seems to be.

Clarity: Yes.

Relation to Prior Work: Yes.

Reproducibility: Yes

Additional Feedback: Please compare the results with more related works. I have read the authors' rebuttal. I would like to upgrade my score as slightly above the acceptance threshold.


Review 2

Summary and Contributions: The paper tackles the problem of improving a normal classification problem by pretraining on a series of potential auxillary tasks. By leveraging the observed sequence of values, the model learns a latent representation through an autoencoder and then improves performance for a small number of datapoints. In particular, the model learns weights on the auxillary task in order to weigh them as "useful" to the classification task of interest. The paper demonstrates the method on the MIMIC-III dataset to show that this method outperforms baselines when the full dataset is subsampled to 1% or 10% fo the original size. The paper also includes a careful ablation study to guide multi-task learning.

Strengths: Overall, I enjoyed this paper. The idea of the paper is elegant and plausible. The mathematical derivations are clear and well-reasoned. The empirical evaluation demonstrates that in real world settings, the model outperforms baselines in regimes with low data. The ablation studies are thoughtful and shed light on how the model is working. I was also impressed with the thoroughness of the experimental details in the supplementary. Beyond the methodological contribution, the detailed analysis of the auxiliary tasks that most benefit each supervised task would be useful for others doing multi-task learning on the same dataset.

Weaknesses: Not so much weaknesses as areas of additional curiosity: 1) The carry forward data imputation of the dataset may be teaching the auxiliary tasks to predict the most recent value by default. Is this a strength or a weakness? 2) For what patients is this model serving well? One might imagine that this model biases for patients with a lot of observed data in the auxiliary tasks, i.e. longer stays in the ICU. 3) Can we reason about model convergence? How well does the model overcome local optimum? Specifically computing the weights of auxiliary tasks may lead to degenerate solution. How sensitive are the learned weights to dataset noise or bad initialization? --- I read all other reviews and author feedback. As the authors answered my questions sufficiently, I am keeping my score the same.

Correctness: To the best of my knowledge through careful review, the equations appear correct. The analysis of the results is generally sound although I do question the claim (line 212-213) that CoTrain has a higher variance than PreTrain in Table 2.

Clarity: I found the equations easy to reason through. I especially appreciated the algorithm box.

Relation to Prior Work: Yes, the related work section clearly describes prior work and where this work differentiates.

Reproducibility: Yes

Additional Feedback:


Review 3

Summary and Contributions: The authors propose a method that learns a model that can classify a given multi-variate time series. Their learned system is a recurrent neural network (RNN) that maps a multi-variate time series (which represents an individual patient) into a vector, and a classifier that maps that vector to one of C possible classes. (The paper focused on the binary classification task - C=2.) The key points of their proposed method are (1) Learn a vector, lambda, that contains a weight for each of the K individual time series, in the multivariate, K-dimensional, time series, where the weight lambda_k t penalizes errors in the prediction of the k-th time series during the pretraining phase. (2) Pretrain the RNN using a self-supervised learning approach that learns a model that predicts timesteps t+1, …, t+n, given the timesteps 1, …, t of the multi-variate time series; then, use that pretrained model as a starting point for fine-tuning of the RNN and the classifier for the main task. They propose an algorithm to learn the weights of the RNN, the classifier, and the weights. They validate their proposed algorithm in a clinical dataset, where each patient is defined by a 96-dimensional time series. Each of the 96 time series represents the measurement of a variable over time (e.g. blood pressure, creatinine level, etc.). They predicted the labels for 3 binary-tasks (event vs non-event in prediction of mortality, shock, and kidney failure), and compared the performance of their approach with traditional supervised learning, multi-task learning, and pre-training giving equal weight (lambda) to each of the 96 time series. They demonstrate that their method has higher predictive performance (measured by AUC-ROC) than the other methods, being particularly useful when using only a small training set (1% of the total data in their experiments). By analyzing the learned weights, lambda, they identify which features (individual time series) contribute to the final prediction of each of binary tasks.

Strengths: (1) The presented idea is applicable to other tasks of learning a model that can (categorically) classify a multivariate time series. (2) Previous work in other areas of knowledge (NLP, computer vision) has provided ways to use self-supervised learning to learn a good data representation, which can be used to learn an effective classifier. This paper shows empirically that this idea is also applicable to multivariate time series in clinical applications. (3) They propose a way to determine which of the features (univariate time series) are more relevant to learn a data representation that is useful for classification purposes, by assigning a weight to every feature that penalizes errors in the predictions of future steps of every time series, then back-propagating the error on the classification task to update the weights.

Weaknesses: (1) The main concept behind self-supervised learning is to take advantage of large, unlabeled datasets to learn a good representation of data. Then, use this representation to learn a classifier using a labeled dataset. Ideally, the data representation learned during the initial phase will be ‘good enough’ to allow the learning of an accurate classifier with a small labeled dataset. This work, however, used the same dataset for both the self-supervised part and the subsequent classifier-learning part. Also, when training the classifier, they learn a new representation of the data, which might be very different from the one learned in the self-supervised phase. It is not clear what the advantage of using the self-supervised phase is as they seem to use exactly the same data for both phases. The current manuscript should explain why it expects this method to work under these circumstances. (2) The paper claims that using the self-supervised phase has a better performance than not using it, especially when they use a small training dataset (1% of the total data available). However, the experiments performed do not show whether this improvement is due to the combination of self-supervised learning + supervised learning, as claimed, or if the improvement is due to other reasons. In their experiments, the labels are obtained by thresholding the value of one the 96 time signals (e.g. if creatinine > 2 after 48 hours, then the label = 1, and 0 otherwise). Of course, as part of the self-supervised learning phase, they learn to predict the future levels of creatinine. They could use this model to directly predict the creatinine levels for the next 48 hours, and then apply the threshold to the predicted value. This comparison is very important, because maybe the low performance in the supervised setting is due to using the wrong label (a thresholded creatinine level, instead of predicting the creatinine level itself). Note that Figure 2 shows that the performance of the model quickly decreases in the fine-tuning phase, which might suggest that this phase is hurting, rather than helping, the performance. The manuscript did not provide a convincing explanation for this behaviour, beyond attributing it to overfitting due to the small dataset. Note that the pre-training phase uses the same dataset as the fine-tuning phase. The manuscript should explain why there might be overfitting early in the second phase, but not in the first one. (3) Although algorithm 1 is easy to follow, the paper’s associated mathematical derivations are not. In particular, I could not find where the optimization constraints are coming from. To me, they seem like the traditional weight updates of gradient descent, not optimization constraint. The only optimization constraint that is justified in the text is that the sum over all the lambda weights should be equal to 1; however, this constraint does not even appear on their optimization problem formulations in sections 3.1 or 3.2. (4) I could not find the usefulness of section 3.2, since all the derivations shown there are not used at all in their experiments, nor anywhere else. (5) The novelty of the proposed idea is limited for the computer science community. They are applying known ideas in the field of machine learning to a new dataset from the medical domain. = = = = The rebuttal addressed most of my concerns and questions. Eg. they included one extra experiment to answer one of the questions that I had, with results that reinforce the claims made by the manuscript. The rebuttal included an extra bibliographical reference that helped me understand the reasoning for the use of the Lagrangian multipliers during the optimization process.

Correctness: With the information provided, it is difficult to evaluate the correctness of the mathematical derivations in section 3.2. I could not understand why they treat the weight updates of gradient descent as optimization constraints. Also the optimization constraint that the sum of the lambda weights should be equal to 1 is not in the manuscript. Algorithm 1 does look correct, and the paper properly reports the use of hold-out sets (inside cross-validation) to report the results of their approach. However, to attribute the improvement in the prediction performance to the combination of self-supervised learning + supervised learning, the authors need to compare their approach to simply thresholding the ‘self-supervised predictions’. Note that this is equivalent to setting the problem as a traditional regression task.

Clarity: Section 3 of the paper was very difficult to read. The current manuscript uses inconsistent notation, and lacks clarity on why they set the gradient descent updates as optimization constraints. Section 3.2 is especially complicated for the readers. It is not consistent with the use of function arguments, nor sub-indices. For example, Equation 2 shows the weights of the decoder and encoder as sub-indices of the loss function, but Equation 3 uses the sub-index to refer to the validation set, and moved the weights to be arguments of the loss function. Also, the weights of the encoder appear as a function of the parameters lambda, which is weird. The lambda is a parameter of the loss function, not of the weights (e.g. the weights are not a parametric equation that receives lambda as an argument). This section also seems irrelevant to the work and results that they are presenting. === Rebuttal: I'm still confused by Section 3.2, but no other reviewer seemed to have issues with it. I hope the authors can revise, to address my concerns.

Relation to Prior Work: Yes, they have a section mentioning related work on multi-task learning and task selection, meta learning and patient representation learning. They mention minor differences between the work presented in their manuscript and the one presented in other research articles.

Reproducibility: Yes

Additional Feedback: This paper presents an interesting application from the clinical point of view, but with limited novelty from the computational one. Most of my feedback is already in the sections above, but I have some extra question for the authors: (1) What is the relevance of section 3.2 in your manuscript? I couldn’t find where you use the derivations presented there. I saw Algorithm 2 in the appendix, but you did not provide any experimental results. (2) Why do you set the weight updates in gradient descent as optimization constraints? Why is the traditional gradient-descent approach not enough, requiring you to construct a Lagrangian? Why is the constraint on lambda not included there? (3) Self-supervised techniques usually take advantage of large, unlabeled datasets to learn data representations that are better than the ones that one could learn by using data from a smaller, labeled dataset. You seem to be using datasets of the same size for both the self-supervised and the fully supervised part. What advantages do you expect to obtain from this setting? I can see the advantage of using multitask learning, but why is doing multitask regression not enough?


Review 4

Summary and Contributions: The authors propose a self-supervised pre-training method which automatically selects and mixes the most suitable task to achieve the best (a priori) known supervised task. The gradient coming from the supervised task is used to weight the different tasks. Tasks are representations of time series of features that are learnt in a self-supervised training scheme.

Strengths: -The authors show empirically that their framework reaches better performance in three classification tasks due to their automatic task selection (all three tasks are for Mimic-III dataset). -The submission introduces a meta-learning approach to combine multi-task with transfer learning.

Weaknesses: -Although there are interesting and insightful ablation studies, the experiments are only performed on a single dataset Mimic-III. For this dataset, the motivation that joint pre-training is inefficient does not really hold as in my opinion the dataset is not large enough to reach the computational limits of today’s systems. -If I understand the framework correctly, the whole pipeline needs to be retrained for every new supervised task (except for the transfer learning experiment which comes with a performance drop), this seems inefficient.

Correctness: The claims and methods seem to be correct.

Clarity: The paper is well written and understandable. The motivation for the submission is clear as well.

Relation to Prior Work: -In the related work section, there is a clear discussion on the difference to previous work. -I miss the comparison to other multitask or transfer learning approaches in the experiments. The comparison is only performed to a simple, baseline multi-learning approach.

Reproducibility: Yes

Additional Feedback: A lot (or all) hyper-parameters are given in the submission or the appendix and the code is given as well. What I am missing, is a script on how to train the model and actually reproduce the results. The code is given which is great, but it would be more convenient if a run script would be shared as well. This would make it easier to run their code. Questions for the authors: -From looking at plot 2a, it is not clear to me why the pre-training was stopped at that point. The learning curve did not seem to flatten. More evidence on this would be appreciated. Why did the authors stop after 5000 pre-training steps? -Why is there no comparison to other multi-task or transfer learning methods than the simple co-learning approach? -Why is the full pre-training approach not able to outperform the performance of the supervised subset approach? I would be interested in the authors’ opinion. -Minor comments: -Equation 5 is too wide ----------- I read all reviews and authors' responses. As the authors answered my questions sufficinetly, I keep my score

[Author Response · NeurIPS 2020]

We thank the reviewers for valuable and timely comments. We'd like to first emphasize the challenges and contributions:

• Finding the most relevant auxiliary forecasting tasks for pre-training and knowledge transferring to a given primary
clinical outcome prediction task is challenging in that it is by nature a combinatorial optimization problem. Our
paper solves this problem by building a new connection between multitask learning and transfer learning within the
framework of meta-learning in an end-to-end fashion, which is new and has not been addressed before.

• We develop a novel two-loop optimization framework where the learned representation from pre-training is guided by
the generalization performance of the target task rather than being model agnostic. We present an efficient first-order
learning algorithm which learns the task weights end-to-end among up to hundreds of auxiliary time-series tasks.

• Extensive experiments with thorough generalization analysis demonstrates the superior predictive performance of
our method and its robustness, interpretability, and generalization capability to unseen target tasks.

**[Reviewer #3] I.** Self-supervised Training (ST) exploits the abundant training signals that can be easily obtained from
the data itself as augmentation to human labels. On the same dataset, it is often that only a small portion of the data has
human labels especially for medical data, so the model can be first trained against these cheap signals via ST to get
familiar with the structure of the data itself and then fine-tuned against expensive human labels. In our case, 1% of the
time-series have human labels, but ST is conducted using all the patient time-series.

**II.** The hyperparameter $\lambda$ is normalized by a softmax layer preserving $\sum_i \lambda_i = 1$ and optimized together with the
model parameters in a two-loop learning process. The outer loop directly optimizes $\lambda$ by minimizing the validation loss
from the target task via hyper-gradient. This effectively change the hypothesis space from which our model parameters
are optimized against the training loss in the inner loop. Section 3.2 explains how to calculate this hyper-gradient of $\lambda$
by following the Lagrangian formulation originally used to analyze the back-propagation algorithm (A Theoretical
Framework for BackPropagation, LeCun, 1988), and widely adopted in the literature [14, 15, 35]. Because $\lambda$ determines
the hypothesis space of the model parameters, they implicitly depend on $\lambda$ in Equation 8. Finally, Algorithm 1 is based
on the first order approximation to the Jacobian and Hessian in Equation [10-13].

**III.** The sub-index notation is slightly overloaded to indicate the dependency of the loss function on the parameters and
on the dataset in Equation 2 and 3 explicitly. We would like to further polish the notation to be more consistent. Figure
2 plots the AUC on the validation dataset. It shows the learned representation gradually improves the generalization
of the target task during pretraining stage. The decay in the finetune stage is due to overfitting since the training loss
continues to improve for all compared methods.

**IV.** We also present the suggested experiment where **predicting the creatinine levels for the next 48 hours is used
for pretraining** and then thresholding is applied for predicting the target task (kidney failure). From 1%, 10%, and even
100% data, **this method achieves 0.735(0.009), 0.833(0.017), and 0.897(0.012)**, which are worse than our approach
due to the fact that the target task can receive additional benefits from other correlated trajectory predictions.

**[Reviewer #1 and #4] I.** In addition to AUC, **Average Precision (AP)** is reported in the Appendix, providing a
quantitative evaluation of the relation between precision and recall(sensitivity). **II.** Most recent multitask learning
algorithms focus either on the adaptability of trained models into new multitask learning settings or finding a mixing
strategy specific to a few NLP tasks (**up to 8**) in [25, 26, 27]. By contrast, we have up to 100 tasks, and we further
compare to the two-stage task selection work of [25] on the mortality task. From 1% and 10% data, **[25] achieves
0.805(0.001), 0.855(0.012)**, which are worse than our approach due to the separation of the task selection and task
weighting phases. To the best of our knowledge, we are the first to handle the automatic task selection problem in the
time series domain end-to-end by the time of writing.

**[Reviewer #2] I.** In the auxiliary tasks, the model is trained only using the true values as the targets instead of the
imputed values. Because the event sequence of each patient is restricted to a window of 48 hours into both the future
and the past history, the bias towards longer stays can be alleviated. **II.** Across the 10 different training data splits, 15
selected tasks for mortality prediction consistently rank within top 20 of all the tasks, which are annotated in Figure 2b.
Moreover, the generalization analysis is designed for verifying the robustness of the learned task weights $\lambda$. $\lambda$ learned
from the mortality target task can be directly used for shock and kidney failure prediction since mortality is a more
general endpoint than specific dysfunctions shown in Table 3.

**[Reviewer #4] I.** 5,000 iterations in Figure 2a are selected for the pretraining phase as the loss (MSE) of the auxiliary
tasks converges and the validation performance of the auxiliary tasks reaches the peak. **II.** MIMIC-III is the largest
open medical dataset, and as shown in the generalization analysis, the learned task weights can be generalized to unseen
similar tasks without being retrained every time. The full-pretraining approach learns a representation that is agnostic
to the target task. The useful information from relevant auxiliary tasks could be overwhelmed by noisy signals from
unrelated tasks. In contrast, for AutoSelect, the task weights are guided by the generalization performance of the target
task so that the learned representation from the pretraining stage is task aware. In spirits, this process is similar to PCA
where the selected auxiliary tasks can be treated as the 'principle components'. This is also verified by Table 3a where
'Pretrain (Top)' is much better than 'Pretrain (Down)'.

[Meta-Review · NeurIPS 2020]

The paper addresses an important problem in ML with EMR data: how to make the most of this richly structured dataset, where the actual labels of interest might be sparse. The proposed approach is elegant and well-executed, building on the strengths of EMR data in a way that I believe will be an inspiration for much future work. While the experiments are satisfactory, reviewers suggested several important new experiments and experimental evaluations, and I trust the authors will include these (some already present in the authors' response) in the revised version of the paper.